# Real World Estimate of Vaccination Protection in Individuals Hospitalized for COVID-19

**DOI:** 10.3390/vaccines10040550

**Published:** 2022-04-01

**Authors:** Antonella d’Arminio Monforte, Alessandro Tavelli, Sara De Benedittis, Francesca Bai, Camilla Tincati, Lidia Gazzola, Ottavia Viganò, Marina Allegrini, Debora Mondatore, Daniele Tesoro, Diletta Barbanotti, Giovanni Mulé, Roberto Castoldi, Anna De Bona, Teresa Bini, Davide Chiumello, Stefano Centanni, Sabrina Passarella, Nicola Orfeo, Giulia Marchetti, Alessandro Cozzi-Lepri

**Affiliations:** 1Unit of Infectious Diseases ASST Santi Paolo e Carlo, Department of Health Sciences, University of Milan, 20142 Milan, Italy; alessandro.tavelli@gmail.com (A.T.); sara.debenedittis@asst-santipaolocarlo.it (S.D.B.); francesca.bai@unimi.it (F.B.); camilla.tincati@unimi.it (C.T.); lidia.gazzola@asst-santipaolocarlo.it (L.G.); ottavia.vigano@asst-santipaolocarlo.it (O.V.); marina.allegrini@unimi.it (M.A.); debora.mondatore@unimi.it (D.M.); daniele.tesoro@unimi.it (D.T.); diletta.barbanotti@unimi.it (D.B.); giovanni.mule@unimi.it (G.M.); roberto.castoldi@unimi.it (R.C.); anna.debona@asst-santipaolocarlo.it (A.D.B.); teresa.bini@asst-santipaolocarlo.it (T.B.); giulia.marchetti@unimi.it (G.M.); 2Intensive Care Unit ASST Santi Paolo e Carlo, Department of Health Sciences, University of Milan, 20142 Milan, Italy; davide.chiumello@asst-santipaolocarlo.it; 3Pneumology Unit ASST Santi Paolo e Carlo, Department of Health Sciences, University of Milan, 20142 Milan, Italy; stefano.centanni@unimi.it; 4Medical Direction ASST Santi Paolo e Carlo, 20142 Milan, Italy; sabrina.passarella@asst-santipaolocarlo.it (S.P.); nicolavincenzo.orfeo@asst-santipaolocarlo.it (N.O.); 5Institute for Global Health, University College London, London NW3 2PF, UK; a.cozzi-lepri@ucl.ac.uk

**Keywords:** COVID-19, age, anti-SARS-CoV-2 vaccination, in-hospital death, mechanical ventilation

## Abstract

Whether vaccination confers a protective effect against progression after hospital admission for COVID-19 remains to be elucidated. Observational study including all the patients admitted to San Paolo Hospital in Milan for COVID-19 in 2021. Previous vaccination was categorized as: none, one dose, full vaccination (two or three doses >14 days before symptoms onset). Data were collected at hospital admission, including demographic and clinical variables, age-unadjusted Charlson Comorbidity index (CCI). The highest intensity of ventilation during hospitalization was registered. The endpoints were in-hospital death (primary) and mechanical ventilation/death (secondary). Survival analysis was conducted by means of Kaplan-Meier curves and Cox regression models. Effect measure modification by age was formally tested. We included 956 patients: 151 (16%) fully vaccinated (18 also third dose), 62 (7%) one dose vaccinated, 743 (78%) unvaccinated. People fully vaccinated were older and suffering from more comorbidities than unvaccinated. By 28 days, the risk of death was of 35.9% (95%CI: 30.1–41.7) in unvaccinated, 41.5% (24.5–58.5) in one dose and 28.4% (18.2–38.5) in fully vaccinated (*p* = 0.63). After controlling for age, ethnicity, CCI and month of admission, fully vaccinated participants showed a risk reduction of 50% for both in-hospital death, AHR 0.50 (95%CI: 0.30–0.84) and for mechanical ventilation or death, AHR 0.49 (95%CI: 0.35–0.69) compared to unvaccinated, regardless of age (interaction *p* > 0.56). Fully vaccinated individuals in whom vaccine failed to keep them out of hospital, appeared to be protected against critical disease or death when compared to non-vaccinated. These data support universal COVID-19 vaccination.

## 1. Introduction

COVID-19 pandemic does not seem to be coming to an end worldwide any time soon and most countries in Europe are in the middle of a fourth wave of infections with a new variant of concern (VoCs) [1,2]. There are several circumstances which are determining the persistence of this pandemic: first, the number of individuals who are still not vaccinated. Indeed, at least in the USA and Europe, where notification of cases is closely monitored, the number of cases at country level is inversely proportional to the percent of vaccinated individuals: the lowest is this percentage the highest is the number of new COVID-19 cases and deaths [3,4].

A second possible cause of persistence of the pandemic is that vaccines might lose their efficacy against the risk of infection or severe COVID-19 disease with longer months from the date of infusion. Information regarding the durability of protection is limited by the fact that all the trials on anti-SARS-CoV-2 vaccines have a short follow up [4,5,6,7,8,9]; on the other hand, it was inevitable as in a pandemic setting, it is mandatory to urgently stop the virus to spread and to reduce the associated risk of death without waiting for the 2- or 3-year vaccine efficacy data [10].

The role of VoCs in maintaining the spread of the virus also needs to be better determined: it is well known that RNA viruses are able to mutate quickly in favor of VoCs with higher contagion rate, but it remains unclear whether infections with the newer VoC are associated with a more severe disease [11,12].

Given these observations, the aims of our study are to describe the frequency of vaccination at hospital admission in individuals who required hospitalization because of symptomatic COVID-19 disease over the year 2021. We also aim to describe the clinical presentation and course of the disease in fully vaccinated participants as compared to those who were partially vaccinated and the non-vaccinated. Our primary objective was to ascertain whether and to what extent fully vaccinated individuals have a reduced risk of death even when they require hospitalization because of COVID-19. 

## 2. Materials and Methods

### 2.1. Study Design

Prospective observational cohort study, including all the patients who were admitted to the San Paolo Hospital in Milan with a diagnosis of symptomatic SARS-CoV-2 infection between 1 January 2021, the date of the introduction of the anti-SARS-CoV-2 vaccine campaign in Italy, and 31 December 2021.

### 2.2. Subjects and Methods

Patients were included in the analyses if they satisfied the following inclusion criteria: (i) a confirmed diagnosis of symptomatic SARS-CoV-2 infection by RT-PCR on naso-pharyngeal or oro-pharyngeal or broncho-alveolar swab specimens; (ii) age ≥ 18 years and (iii) hospitalization at San Paolo Hospital, over the period January-December 2021. Patients who died in the emergency room within 24 h from arrival to hospital and those who did not need to be hospitalized, as asymptomatic or pauci-symptomatic, were not included in these analyses. 

Data at hospital admission were entered into an electronic database, including: age; sex; ethnicity (Caucasian, Latin/Hispanic, Black, Asian, other); number and details on ongoing or previous comorbidities, i.e., obesity (defined as Body Mass Index –BMI-> 30 kg/m^2^); cancer; cerebro-vascular disease; chronic kidney failure (CKD); congestive heart failure; connective tissue disease; chronic obstructive pulmonary disease (COPD); dementia; diabetes mellitus; end-stage liver disease (ESLD); hematologic disease; HIV/AIDS; arterial hypertension; age-unadjusted Charlson comorbidity index (CCI) [13]; date of onset of symptoms; respiratory rate (RR); pulse oxymetry and oxygen saturation percent (SO2); baseline PaO2, PaO2/FiO2; laboratory parameters (haemoglobin, white blood cells, lymphocytes, platelets, ALT, C-reactive protein -CRP-, procalcitonin, d-dimer, lactic dehydrogenase -LDH-, creatinine, creatine phosphokinase).

The highest intensity of ventilation during hospitalization and date of occurrence was recorded as: no need of ventilation; low and high flow supplemental oxygen by nasal cannula or face mask; continuous positive airway pressure device (cPAP); mechanical non-invasive (NIV) or invasive (IV) ventilation. 

Use of monoclonal antibodies prior to admission was registered. Date of start and stop of remdesivir, dexamethasone, immunomodulatory agents (tocilizumab, sarilumab, other) and low molecular weight heparin post the date of admission were also collected. 

Total number of doses, date of last dose and type of vaccine (mRNA vaccines: BNT162b2—Pfizer- or BNT mRNA-1273-Moderna, ChAdOx1-Astra Zeneca) were also recorded. We excluded from the analyses six patients who received Ad26.COV2-S vaccine because of its different schedule of vaccination.

### 2.3. Statistical Analyses

Baseline characteristics of the participants, assessed at time of hospitalization, were compared after stratification by anti-SARS-CoV2-vaccination history (our main exposure of interest). Continuous variables were expressed as median (IQR) and compared by Mann-Whitney U test. Categorical variables were expressed as numbers and percentages and compared by χ^2^ test or Fisher’s exact test by treatment strategy. If a signal for a difference in the level of biomarkers measured at hospital admission by vaccination status was detected, a multivariable ANOVA analysis was conducted which controlled for potential confounders (Appendix A).

In the main analyses, anti-SARS-CoV-2 vaccination was coded as: no vaccination; full vaccination cycle (second dose of vaccination dated at least 14-days before onset of symptoms; third dose), partial vaccination cycle (second dose of vaccination missing or dated less than 14-days before onset of symptoms). Since the risk of the in-hospital outcomes was similar when comparing the unvaccinated with the partially vaccinated group some of the analyses were conducted using a binary exposure variable bumping together the non-vaccinated and the partially vaccinated vs. fully vaccinated. 

The monthly prevalence of the rate of vaccination at the date of hospital admission was calculated and plotted in a bar chart over the period January–December 2021. 

The primary endpoint was time from the date of admission to in-hospital death. The secondary endpoint was time from the date of admission to initiation of mechanical ventilation (cPAP, and/or NIV or IV) or in-hospital death. 

The association between our main exposure anti-SARS-CoV-2 vaccination and the risk of outcomes was evaluated by means of Cox regression models analysis, adjusting for factors identified as confounders (age, age-unadjusted CCI, ethnicity, month of admission) (Appendix A).

We performed several sensitivity analyses (i) after restricting to participants with a PaO2/FiO2 at admission < 300 mmHg, (ii) among the fully vaccinated, only those who have received 2 doses. We also conducted an adjusted analysis by controlling for immunocompetence (concomitant diagnosis of cancer or HIV) instead of controlling for CCI.

We finally evaluated whether age and calendar year of admission were effect measure modifiers for the relationship between history of vaccination and the risk of outcomes. Participants were grouped in age strata (19–70, 71–80 and 81+ years) using pre-specified cut-offs chosen based on a previous work on this same cohort investigating age as a predictor of death [14]. Calendar period was divided in the months January–June 2021 vs. July–December 2021 under the hypothesis that the Wuhan D614G strain was circulating in the early part of the year which was replaced by the delta VoC starting from July. Potential effect measure modification was formally tested by including an interaction term in the Cox regression model. We also investigated whether the effect of vaccination was different according to age (using 10 years age bands) by fitting a logistic regression and model predictions with 95% CI by age and vaccination status after controlling for other confounders were shown.

### 2.4. Ethics Consideration

The study was approved by Ethic Committee Area 1, Milan (2020/ST/049 and 2020/ST/049_BIS, 11 March 2020). Informed consent was obtained whenever possible.

## 3. Results

A total of 956 patients were hospitalized at San Paolo Hospital over the time frame of the study. Of these, 151 (16%) had already completed a full cycle of anti-SARS-CoV-2 vaccine (18 of them had received also the third dose), 62 (7%) had received only one dose of vaccine, and 743 (78%) had not received any vaccination at all. Demographic and clinical characteristics according to vaccination status are shown in Table 1.

Fully vaccinated participants were older (median 80-IQR: 69–74-years old, vs. 76-IQR: 62–84- and 67-IQR: 54–80; *p* < 0.001 in those receiving one dose or no doses of vaccine) and with a larger number of comorbidities (mean age-unadjusted Charlson Comorbidity Index: 1.6-SD 1.7-, vs. 0.7-SD 1.0, vs. 0.9-SD 1.8; *p* < 0.001, Table 1). Of note the median number of days between the end of the full vaccination to symptoms onset was of 176 days (IQR:131–200).

Of the laboratory parameters measured at the time of hospital admission (Table 2), we found that LDH was higher in the unvaccinated individuals vs. fully vaccinated, after adjusting for age, age- unadjusted Charlson index and month of admission. By comparing unadjusted median levels, there was a signal also for hemoglobin, serum creatinine and C reactive protein to be higher in the vaccinated but the difference was largely explained by confounding factors (data not shown). 

At the time of initiation of the vaccination cycle, the anti-SARS-CoV-2 vaccine most frequently used was BNT162b2—Pfizer—(148 cases, 69.5%), followed by ChAdOx1—Astra Zeneca (*n* = 43, 20.2%), mRNA-1273—Moderna (*n* = 22, 10.3%). Out of the 151 participants who received full vaccination 18 (12%) had received a booster dose of BNT162b2 (*n* = 14, 9.3%) or mRNA-1273 (*n* = 4, 2.7%) vaccine. Details of the type of vaccine used according to number of doses received are shown in Table 1. 

The prevalence of participants who received a full cycle of anti-SARS-CoV-2 vaccine before hospitalization for COVID-19 increased by calendar month of hospital admission ranging from 13% of the 120 cases hospitalized in April, to 62% of those (*N* = 113) hospitalized in December (Figure 1). 

Anti-SARS-CoV-2 monoclonal antibodies were given prior to hospitalization more frequently to fully vaccinated participants than to unvaccinated ones (*p* < 0.001). After admission, remdesivir was given more frequently to unvaccinated participants (*p* < 0.001) (Table 3).

Regarding the highest level of respiratory support needed during hospitalization, high flow oxygen therapy was more extensively used among fully vaccinated participants; there was no evidence for a difference according to history of vaccination for any other type of support (Table 4). 

Over a median follow-up of 12 days (IQR: 7–20), the primary endpoint of in-hospital death, occurred in 203 (21%) of participants, with no evidence for a difference by vaccination status: 21% of patients unvaccinated, 29% of vaccinated with one dose and 21% of the fully vaccinated participants who died in hospital (*p* = 0.18). There was no evidence for a difference by vaccination status also for the risk of developing the secondary endpoint, i.e., the need of mechanical ventilation (cPAP or NIV or IV) or occurrence of in-hospital death, which overall occurred in 490 (51%) participants (Table 4). 

The Kaplan-Meier probabilities of in-hospital death and mechanical ventilation or death are shown in Figure 2A,B. By 28 days, the risk of death was of 35.9% (95% CI: 30.1–41.7) in unvaccinated, 41.5% (24.5–58.5) in vaccinated with one dose and 28.4% (18.2–38.5) in vaccinated with two or three doses (log-rank test *p* = 0.63). Similarly, by 14 days the probability of mechanical ventilation or death were of 49.8% (95% CI:46.2–53.4%) in the unvaccinated, 46.8% (95% CI:34.4–59.2%) in the partially vaccinated and 41.7% (95% CI:33.9–49.6) in the fully vaccinated (log-rank test *p* = 0.17). In a Kaplan Meier analysis weighted for confounders, the day 28 cumulative risk of the primary endpoint was considerably lower than that seen in the other groups with a log-rank test *p*-value approaching significance (Figure 2C, *p* = 0.14) in fully vaccinated. The curves of one dose vs no doses vaccine overlapped and the results from the adjusted Cox regression confirmed no evidence for a difference on both primary and secondary endpoints between these groups, primary endpoint AHR: 1.04 (95% CI: 0.63, 1.71); secondary endpoint: AHR 0.83 (95% CI: 0.57, 1.19).

Relative hazards of in-hospital death (panel A) and of the secondary outcome of mechanical ventilation or death (panel B) are shown in Table 5 from fitting an univariable model and after controlling for age, age-unadjusted Charlson index, ethnicity and month of enrolment. In the adjusted analysis, using the unvaccinated as the comparator group, fully vaccinated participants showed a risk reduction of 50% for both the rate of in-hospital death, AHR 0.50 (95% CI: 0.30–0.84) and for the composite endpoint of mechanical ventilation or death, AHR 0.49 (95% CI: 0.35–0.69). Results were similar after controlling for immune-competence instead of CCI (Appendix A). Of note, even after controlling for confounding factors, there was no evidence for a difference in the hazards of developing the endpoints comparing participants who received only one dose of anti-SARS-CoV-2 vaccine vs. those unvaccinated [primary endpoint AHR: 1.04 (95% CI: 0.63, 1.71); secondary endpoint: AHR 0.83 (95% CI: 0.57, 1.19)].

The magnitude and significance of the effect of full vaccination on the risk of death was similar after removing participants who received the booster dose (AHR: 0.55; 95% CI: 0.33–0.93 vs. unvaccinated; *p* = 0.02). Results were similar after restricting the analysis to participants who completed the full vaccination cycle more than 180 days prior to admission and regardless of whether they had the third dose (Table 5, panels c and d). We also directly investigated the association between length of time elapsed since the date of last dose received in the 2+ doses recipients and data carried weak evidence for an association (AHR = 0.88, 95% CI:0.56–1.38, *p* = 0.57, Appendix A).

Results were also similar in those who started the vaccination with BNT162b2 instead of other types of vaccines (Appendix A) and after restricting the analyses to those with severe disease at admission (PaO2/FiO2 < 300 mmHg) (Appendix A). 

Finally, we evaluated the potential effect measure modification by age. From plotting the predicted probability of developing the primary endpoint obtained from a logistic regression approximating the hazards with the odds, there was a signal for the effect of vaccination to be bigger in participants with older ages (e.g., 41% vs. 59% in the 90 years old vs. 15% vs 18% in the 70 years old) comparing fully vaccinated with unvaccinated. In contrast, we obtained parallel predictions by age strata for the secondary composite endpoint (Figure 3).

However, when we formally tested for interaction in the Cox regression model, for both the primary and secondary endpoint, we found no evidence that the difference in risk varied by age strata and by calendar period of hospital admission (Figure 4, interaction *p*-value > 0.05).

## 4. Discussion

Our analysis shows that among participants hospitalized for COVID-19 (some despite vaccination), the risk of in-hospital death after admission in fully anti-SARS-CoV-2 vaccinated was half of that seen for unvaccinated individuals. Thus, even in people in whom, for whatever reason, vaccination was not sufficient to keep them out of hospital, once they were admitted, they showed a more favorable prognosis than those who were admitted and had not received the anti-SARS-CoV-2 vaccine. Although reasons for not been vaccinated at time of hospitalization are unknown, it is important to note that a key possible reason was lack of availability of the vaccine at the time of infection. 

People hospitalized after full vaccination were elderly (median age 80 years) and with a high burden of comorbidities. This suggests that vaccination appears to offer less protection from COVID-19 in patients with an impaired immune-system due to age-related immune senescence [15], and/or in those with pre-existing comorbidities which can quickly progress due to concomitant infection with SARS-CoV-2 virus. Older participants also showed longer time from the date of the vaccination with a second dose to hospitalization, (approximately 6 months on average) which may have contributed to the waning of protection. This might also be explained by the fact that older people in the Lombardy region received their second dose earlier in the year which has resulted in some of them to have lost protection while waiting for the booster dose. We also observed that the prevalence of hospitalized fully vaccinated participants at hospital admission was much higher in recent months as compared to the beginning of the year. This might be due to extended availability of vaccination for all the people independently of age and comorbidities, only later in the year. 

Hemoglobin levels, serum creatinine and C reactive protein were more frequently altered in vaccinated individuals as compared to the unvaccinated: this was largely explained by older age and concomitant diseases. On the other hand, LDH serum levels, related to lung damage, were more frequently altered in unvaccinated individuals independently of other factors, even if the degree of respiratory insufficiency at admission was similar across the groups of participants. 

The prevalence of in-hospital death was very high, around 20%, in our setting. In a systematic review of Macedo and others [16] based on more than 13,000 hospitalized patients, the rate of fatality was 17%, (12% in non-critical COVID-19, and 41% among patients with critical disease). In our setting participants with critical disease (defined by a PaO2/FiO2 < 100 mmHg at admission) had a fatality rate of 49% (95% CI: 41–59) (data not shown). It has been already observed that fatality rates are very high in Lombardy [17] and several explanations have supported this finding: high pollution, highly industrialized country, high population density [18]. 

We adjusted the analyses for known confounders, i.e., factors affecting both vaccination and in-hospital death: age, ethnicity, the complexity of comorbidities as coded by age-unadjusted Charlson Comorbidity Index and month of admission [14,19,20].

In univariable analyses, the rate of disease progression was similar by vaccination status. However, an important difference was masked by confounding factors. Indeed, after controlling for age, complexity of comorbidities and month of admission, fully vaccinated patients showed a 50% reduction in the risk of in-hospital death and of progression to mechanical ventilation or death compared to unvaccinated. In contrast, there was no evidence for a difference in these risks when comparing unvaccinated and partially vaccinated participants. It has been shown that vaccination with a third dose increases protection from infection and progression to severe disease [21,22]. In our setting, results were similar after excluding participants who had received the third dose. It should be noted that only a few participants had received the booster dose because the third dose campaign started in the autumn 2021. As a consequence, the potential additional effect of the booster dose could not be evaluated. 

A second key finding was the lack of interaction between age, calendar period (used as proxy of currently circulating VoC) [23] and vaccination status for both our endpoints. This was particularly evident for the risk of mechanical ventilation or death for which the protecting effect of vaccination was the same regardless of age. This is important as it shows that even the older fragile population who developed COVID-19 despite vaccination were protected from further complications after admission. Indeed, the effect of full vaccination tended to be even greater for those aged 60 or above. Possible reasons of protection from critical disease or death despite COVID-19 occurrence after vaccination might be the persistence of cellular response to the virus, as demonstrated by other authors in the HIV setting [24]. The clinical efficacy of the cellular response overall and along the elapsing time after vaccination and whether some protective immunity persists and for how long remains unclear. Of note, our data are inconclusive regarding a possible association between time elapsing from the last vaccine dose and level of vaccine protection against clinical outcome after hospitalization.

These data, together with those showing an absent/reduced spread of SARS-CoV-2 in vaccinated people support the importance of recommending anti-SARS-CoV-2 vaccination at every age. 

To our knowledge there are few reports on the outcome of vaccinated individuals hospitalized with COVID-19. Tenforde et al. [25] reported findings similar to ours in a different setting of a case-control study from 21 US hospitals, demonstrating a decreased likelihood of having been vaccinated among hospitalized patients with SARS-CoV-2 who died or underwent invasive mechanical vaccination. Whittaker et al. [26] analyzed the Norway national registry and demonstrated that fully vaccinated individuals showed a shorter length of in-hospital stay and a lower risk of ICU admission compared to unvaccinated, but no differences on in-hospital death, that accounted for 13% of their patients.

We did not analyze separately the likelihood of ICU stay according to vaccination status as several unmeasured variables might have influenced this occurrence, due to the shortage of ICU beds particularly in the periods of overcrowding hospitalized patients. Taken all together, these data support the importance of recommending anti-SARS-CoV-2 vaccination at every age.

Our analyses have several limitations. First of all, we identified at least one unmeasured confounding factor which is patients’ SARS-CoV-2 serology results or previous disease. COVID-19 survivors in Italy have a reduced chance to be vaccinated (they need to wait >6 months before being able to receive a jab) and previous infection with SARS-CoV2 has been associated with a reduced risk of progression to severe disease [27]. Therefore, we cannot rule out unmeasured confounding bias. However, this means that a proportion of the unvaccinated might have been survivors and therefore the magnitude of the protection conferred by the vaccine could have been even underestimated in our analysis not controlling for serology. In addition, we do not have any information on the specific VoC harbored by our participants and, although non consistently on all studies, it has been shown that the newly circulating variants, in particular Omicron, are associated with a reduced risk of developing severe disease [28,29]. We tried to address this hypothesis by stratifying the analysis according to calendar period of admission (under the assumption that the Delta VoC would be more prevalent after July) and we did not find statistical evidence for an interaction between calendar period and exposure to vaccination. However, according to our assumptions (Appendix A) the backdoor confounding or pathway going through VoC was already blocked by controlling for age and calendar month of admission. 

Furthermore, we also had no information on level of socioeconomic deprivation, which is considered as an important confounding factor in the estimates of real-word effectiveness of vaccination. However, our assumption is that the rate of vaccination was not modified by socio-economic indicators as in Italy vaccination campaign was spread across the whole population and access to care is free of charge. Lastly, we did not have sufficient data to evaluate the protective effect of the booster dose which was seen in other settings [30,31], due to the introduction of the booster dose only in mid-autumn 2021 in Italy. 

## 5. Conclusions

In conclusion, our data confirm that in patients who were admitted to hospital (some despite being vaccinated) administration of two doses of anti-SARS-CoV-2 vaccine can re-duce the risk of in-hospital death and of the need for mechanical vaccination by a remarkable one half. We hereby confirm also that patients with the highest risk of clinical progression after hospital admission are those older and those with comorbidities and that there was no evidence that the protection from full vaccination varied by age. All these findings together further support the need for universal vaccination, to reduce the rate of infections and protect the more fragile an older individual from being hospitalized and reduce mortality in those who are admitted to hospital.

## Figures and Tables

**Figure 1 vaccines-10-00550-f001:**
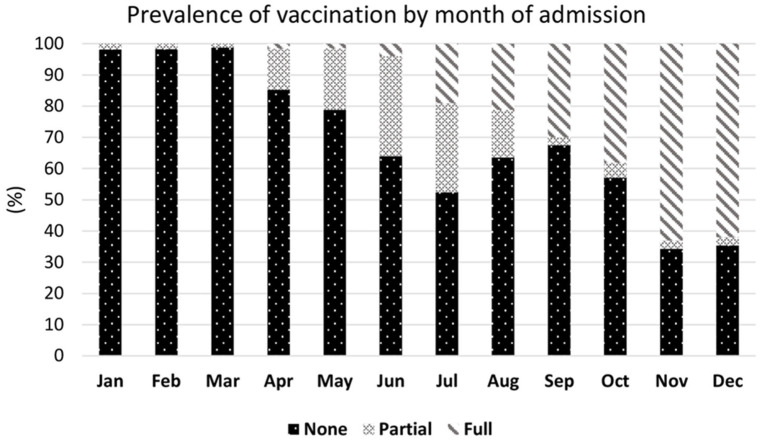
Prevalence of vaccination by month of admission 2021.

**Figure 2 vaccines-10-00550-f002:**
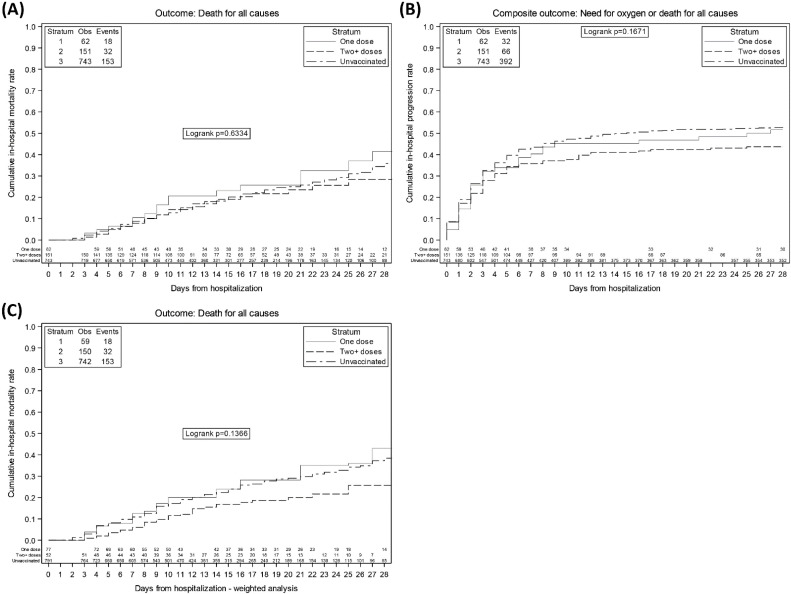
Kaplan Meier curves of: (Panel **A**) in-hospital death, (Panel **B**) need of mechanical ventilation or death, according to vaccination status, (Panel **C**) weighted for confounders.

**Figure 3 vaccines-10-00550-f003:**
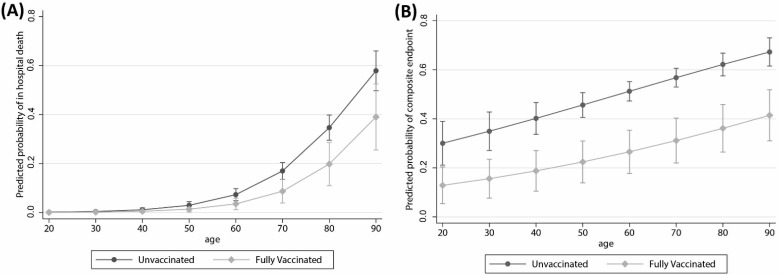
Adjusted predictors of (**A**) in-hospital death and (**B**) mechanical ventilation or death according to fully vaccination status and age.

**Figure 4 vaccines-10-00550-f004:**
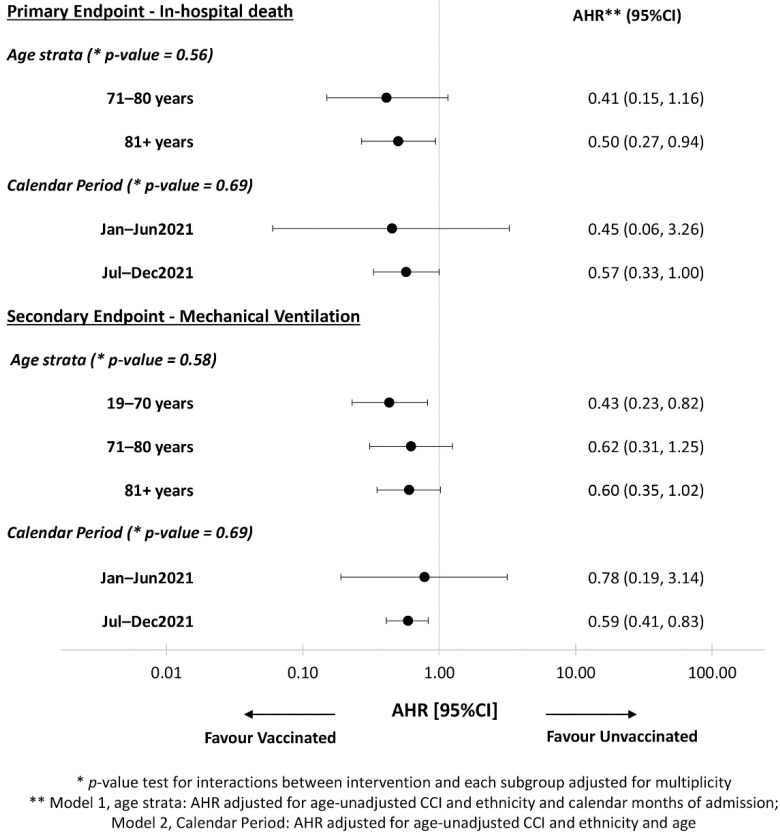
Adjusted risk of primary and secondary endpoints in relation of age by forest plot.

**Table 1 vaccines-10-00550-t001:** Main characteristics at enrolment by vaccination group.

	Anti COVID-19 Vaccination				
Characteristics	None	1 Dose	2+ Doses	*p*-Value *	Total
	*N* = 743	*N* = 62	*N* = 151		*N* = 956
**Age, years**				<0.001	
Median (IQR)	67 (54, 80)	76 (62, 84)	80 (69, 84)		71 (56, 81)
**Gender, *n*(%)**				0.994	
Female	297 (40.0%)	25 (40.3%)	61 (40.4%)		383 (40.1%)
**Ethnicity, *n*(%)**				0.056	
Caucasian	650 (87.6%)	50 (82.0%)	145 (96.7%)		845 (88.7%)
Black	1 (0.1%)	0 (0.0%)	0 (0.0%)		1 (0.1%)
Asian	17 (2.3%)	2 (3.3%)	0 (0.0%)		19 (2.0%)
Ispanic	28 (3.8%)	2 (3.3%)	1 (0.7%)		31 (3.3%)
Other	0 (0.0%)	0 (0.0%)	0 (0.0%)		0 (0.0%)
**Comorbidity, *n*(%)**					
Obesity	100 (45.5%)	3 (42.9%)	16 (28.6%)	0.074	119 (42.0%)
Cancer	52 (7.0%)	5 (8.1%)	29 (19.2%)	<0.001	86 (9.0%)
Cerebro-vascular disease	39 (5.2%)	2 (3.2%)	13 (8.6%)	0.184	54 (5.6%)
Chronic kidney failure	51 (6.9%)	4 (6.5%)	22 (14.6%)	0.006	77 (8.1%)
Congestive failure	21 (2.8%)	0 (0.0%)	2 (1.3%)	0.242	23 (2.4%)
Connective tissue disease	0 (0.0%)	0 (0.0%)	0 (0.0%)		0 (0.0%)
Chronic respiratory disease	59 (7.9%)	6 (9.7%)	28 (18.5%)	<0.001	93 (9.7%)
Dementia	44 (5.9%)	9 (14.5%)	15 (9.9%)	0.014	68 (7.1%)
Diabetes mellitus	145 (19.5%)	9 (14.5%)	36 (23.8%)	0.264	190 (19.9%)
End stage liver disease	7 (0.9%)	0 (0.0%)	1 (0.7%)	0.713	8 (0.8%)
Hematologic disease	5 (0.7%)	0 (0.0%)	5 (3.3%)	0.010	10 (1.0%)
HIV/AIDS	7 (0.9%)	0 (0.0%)	2 (1.3%)	0.662	9 (0.9%)
Arterial hypertension	324 (43.6%)	29 (46.8%)	86 (57.0%)	0.011	439 (45.9%)
Age-unadjusted Charlson Comorbidity Index, mean (SD)	0.9 (1.8)	0.7 (1.0)	1.6 (1.7)	<0.001	1.0 (1.8)
**PO2**				0.988	
Median (IQR)	70 (61, 83)	70 (62, 86)	70 (61, 83)		
**Baseline PaO2/FiO2, mmHg**				0.270	
Median (IQR)	300 (243, 348)	314 (267, 354)	309 (250, 357)		302 (248, 348)
0–250, *n*(%)	183 (27.0%)	12 (22.2%)	33 (25.0%)	0.696	228 (26.4%)
0–150, *n*(%)	57 (8.4%)	4 (7.4%)	14 (10.6%)	0.671	75 (8.7%)
**FiO2**				0.331	
Median (IQR)	0.2 (0.2, 0.3)	0.2 (0.2, 0.2)	0.2 (0.2, 0.3)		0.2 (0.2, 0.3)
**Main delays**					
Days from symptoms onset to hospitalization, median(IQR)	5 (2, 8)	5 (3, 6)	4 (2, 7)	0.602	5 (2, 8)
Days from last vaccine dose to symptoms onset, median (IQR)		7 (2, 18)	176 (131, 200)	<0.001	157 (23, 203)
**Type of vaccine:**					
BNT162b2		37 (60.0%)	108 (71.5%)		145 (68.0%)
mRNA-1273		12 (19.0%)	10 (6.6%)		22 (10.3%)
ChAdOx1		13 (21.0%)	25 (16.6%)		38 (18.0%)
mRNA-mRNA			3 (2.0%)		3 (1.4%)

* Chi-square or Mann-Whitney test as appropriate. Among the 18 patients receiving 3 doses of vaccine: 14 BNT162b2 and 4 mRNA-1273.

**Table 2 vaccines-10-00550-t002:** Basal laboratory parameters by vaccination group.

	Anti-COVID-19 Vaccination				
Blood Tests	None	1 Dose	2+ Doses	*p*-Value *	Total
	*N* = 473	*N* = 47	*N* = 145		*N* = 665
**Markers, Median (IQR)**					
Haemoglobin, g/dL	13.5 (12.1, 14.7)	13.7 (11.7, 15.0)	12.6 (11.0, 14.1)	0.002	13.2 (11.8, 14.5)
White Blood Cells, g/dL	6.8 (4.7, 9.4)	6.1 (4.7, 9.7)	6.5 (4.9, 8.8)	0.850	6.7 (4.8, 9.3)
Lymphocytes, 10^3^/mm^3^	1.0 (0.7, 1.4)	1.2 (0.8, 2.1)	0.9 (0.6, 1.4)	0.018	1.0 (0.7, 1.5)
Alanine amino-transferase (ALT), U/L	29.0 (19.0, 49.0)	25.0 (15.5, 44.0)	23.0 (17.0, 41.0)	0.016	28.0 (18.0, 46.0)
Creatine Kinase, U/L	83.0 (51.0, 180.0)	93.0 (44.0, 157.0)	81.0 (44.0, 175.0)	0.866	84.5 (49.5, 179.5)
Creatinine, mg/dL	0.8 (0.7, 1.1)	0.9 (0.7, 1.4)	1.0 (0.8, 1.4)	<0.001	0.9 (0.7, 1.2)
D-dimer, mg/L	322.0 (217.0, 616.0)	361.5 (149.0, 535.0)	324.5 (204.0, 633.0)	0.709	322.0 (213.0, 626.0)
Lactate dehydrogenase, U/L	288.0 (227.0, 384.0)	286.0 (209.0, 393.0)	250.0 (200.0, 321.0)	0.002	280.0 (219.0, 372.0)
C-reactive protein, mg/dL	49.8 (20.6, 82.1)	36.0 (17.1, 59.6)	55.1 (24.1, 95.6)	0.049	49.7 (21.7, 82.0)
Platelets, 10^3^/mm^3^	196.0 (154.0, 250.0)	176.0 (119.0, 262.0)	189.0 (140.0, 254.0)	0.279	193.0 (150.0, 252.0)
Procalcitonin,	0.2 (0.1, 0.9)	0.2 (0.1, 9.0)	0.2 (0.2, 0.3)	0.538	0.2 (0.1, 1.0)

* Mann-Whitney test.

**Table 3 vaccines-10-00550-t003:** Therapies used by vaccination group.

	Anti COVID-19 Vaccination				
Therapies	None	1 Dose	2+ Doses	*p*-Value *	Total
	*N* = 743	*N* = 62	*N* = 151		*N* = 956
Monoclonal antibodies, *n*(%)	30 (4.0%)	4 (6.5%)	38 (25.2%)	<0.001	72 (7.5%)
Heparin, *n*(%)	579 (77.9%)	45 (72.6%)	110 (72.8%)	0.291	734 (76.8%)
Remdesivir, *n*(%)	186 (25.1%)	18 (29.0%)	5 (3.3%)	<0.001	209 (21.9%)
Glucocorticoids, *n*(%)	586 (78.9%)	44 (71.0%)	112 (74.2%)	0.194	742 (77.6%)
Immunomodulants, *n*(%)	50 (6.7%)	4 (6.5%)	2 (1.3%)	0.035	56 (5.9%)

* Chi-square test.

**Table 4 vaccines-10-00550-t004:** Outcomes by vaccination group.

	Anti COVID-19 Vaccination				
Outcomes	None	1 Dose	2+ Doses	*p*-Value *	Total
	*N* = 743	*N* = 62	*N* = 151		*N* = 956
No Oxygen, *n*(%)	101 (13.6%)	13 (21.0%)	20 (13.2%)	0.263	134 (14.0%)
Low flow Oxygen, *n*(%)	234 (31.5%)	16 (25.8%)	46 (30.5%)	0.642	296 (31.0%)
High flow Oxygen, *n*(%)	50 (6.7%)	8 (12.9%)	29 (19.2%)	<0.001	87 (9.1%)
CPAP, *n*(%)	251 (33.8%)	22 (35.5%)	39 (25.8%)	0.146	312 (32.6%)
NIV, *n*(%)	78 (10.5%)	2 (3.2%)	15 (9.9%)	0.185	95 (9.9%)
Invasive mechanical ventilation (IMV), *n*(%)	29 (3.9%)	1 (1.6%)	2 (1.3%)	0.203	32 (3.3%)
Mechanical ventilation ^$^ or death, *n*(%)	392 (52.8%)	32 (51.6%)	66 (43.7%)	0.146	490 (51.3%)
In-hospital death, *n*(%)	153 (20.6%)	18 (29.0%)	32 (21.2%)	0.185	203 (21.2%)
Follow-up time, days Median (IQR)	12 (7, 20)	14 (7, 24)	14 (9, 21)	0.213	12 (7, 20)
Follow-up time in those who died, days Median (IQR)	10 (6, 16)	10 (7, 21)	10 (6, 15)	0.885	10 (6, 16)

* Chi-square or Mann-Whitney test. ^$^ At least CPAP.

**Table 5 vaccines-10-00550-t005:** Unadjusted and adjusted HR from fitting a standard Cox regression model (a) of in hospital death (primary endpoint); (b) of need of mechanical ventilation (secondary endpoint); (c) of mechanical ventilation, dividing the 2+ doses group in more or less than 180 days prior to admission; (d) of in hospital death after removing those who received 3 doses.

**(a) Unadjusted and adjusted relative hazards of in hospital death**				
	**Unadjusted HR (95% CI)**	***p*-Value**	**Adjusted * HR (95% CI)**	***p*-Value**
Unvaccinated	1		1	
One dose	1.25 (0.76, 2.03)	0.377	1.04 (0.63, 1.71)	0.869
2+ doses	0.96 (0.65, 1.40)	0.825	0.50 (0.30, 0.84)	0.009
**(b) Unadjusted and adjusted relative hazards of composite endpoint**				
	**Unadjusted HR (95% CI)**	***p*-Value**	**Adjusted * HR (95% CI)**	***p*-Value**
Unvaccinated	1			
One dose	0.94 (0.66, 1.35)	0.756	0.83 (0.57, 1.19)	0.303
2+ doses	0.79 (0.60, 1.02)	0.069	0.49 (0.35, 0.69)	<0.001
**(c) Unadjusted and adjusted relative hazards of composite endpoint**				
	**Unadjusted HR (95% CI)**	***p*-Value**	**Adjusted * HR (95% CI)**	***p*-Value**
Unvaccinated	1		1	
One dose	0.94 (0.66, 1.35)	0.756	0.90 (0.62, 1.29)	0.565
2+ doses since less than 180 days	0.82 (0.57, 1.18)	0.274	0.61 (0.40, 0.92)	0.017
2+ doses for more than 180 days	0.76 (0.54, 1.07)	0.120	0.54 (0.36, 0.82)	0.004
**(d) Unadjusted and adjusted relative hazards of in hospital death -after removing those who received 3 doses**				
	**Unadjusted HR (95% CI)**	***p*-Value**	**Adjusted * HR (95% CI)**	***p*-Value**
Unvaccinated	1		1	
One dose	1.24 (0.76, 2.02)	0.389	1.03 (0.63, 1.70)	0.896
2 doses	1.08 (0.73, 1.59)	0.693	0.55 (0.33, 0.92)	0.024

* adjusted for age, ethnicity, age-unadjusted CCI and month of enrolment.

## Data Availability

The data presented in this study are available on request from the corresponding author.

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
