# Peer review of "Real World Estimate of Vaccination Protection in Individuals Hospitalized for COVID-19"

_vaccines, 2022, doi:10.3390/vaccines10040550_

Round 1
Reviewer 1 Report
This is an interesting study elucidating whether COVID vaccination confers a protective effect against progression after hospital admission. The results of the study support the need for universal vaccination I particular among more fragile and older persons. Data are shown in a clear and comprehensive way. I am happy to have reviewed this paper and to my mind it will have an impact on the literature.
Minor corrections
Line 96 desametasoneà dexamethasone
Line 103 anti-SARS Cov2-vaccinationà anti-SARS CoV-2-vaccination
Author Response
This is an interesting study elucidating whether COVID vaccination confers a protective effect against progression after hospital admission. The results of the study support the need for universal vaccination I particular among more fragile and older persons. Data are shown in a clear and comprehensive way. I am happy to have reviewed this paper and to my mind it will have an impact on the literature.
Minor corrections
Line 96 desametasoneà dexamethasone
Line 103 anti-SARS Cov2-vaccinationà anti-SARS CoV-2-vaccination
We thank you for your kind words and review; we have now corrected the spelling mistakes.
Reviewer 2 Report
The study presented is of value regardless that the conclusion of the study sounds for quite a time in media. The scientific approach makes the results solid and may be used as trusty proof. Saying that we believe that the scientific approach was well chosen and the data presented are representative. Having said that I have the following remarks.
Patients under the study were grouped according to the Covid 19 vaccination. I am afraid it is not clearly shown what vaccines they received. Nowadays it is difficult to accept the belief that one dose receivers are immunized, It would be enough to make a statement that they were at a similar risk of Covid 19 as those not vaccinated at all. However, it was not said what time after vaccination they got Covid 19. It should be discussed.
The same is valid (elapsing time) for two doses recipients.
Twenty people who received the third dose should be analyzed separately as they are at least in theory candidates for Covid19 vaccination immunity. If they failed they should be more thoroughly analyzed which should not be difficult in hospitalized patients. This analysis should include the history of illnesses especially those known to hamper immunity. In addition, the patients might be evaluated for the level of antibodies raised by vaccination.
Also, it is important to identify people who are primarily immunocompromised.
It is important to give information about the vaccine used, some vaccines provide so-called full vaccination after one dose only. The term fully vaccinated may be confusing I would prefer to say vaccinated with two doses and with two doses followed by a third booster shot.
Also, it is important to identify people who are primarily immunocompromised. The elapsing time from the second dose should be shown.
The presentation of the results seems to be quite complex which may hamper receiving the message. The following points might be considered by the Authors at revision:
- Lack of vaccination was not intentional but related to the availability of the vaccination program. People admitted to the hospital during one year might be infected by different variants of SARS CoV 2 – it should be discussed whether it was possible or not
- Elapsing time after vaccination play a role but this is not reflected by the data presented in the table form – it should be discussed
- Risk of death and/or MV was analyzed considering confounding factors that play a significant role, it should be discussed
I believe that the paper might be worthy for publication if presented in a shorter form focusing on the data contributing to the message presented in the conclusions.
Linguistical errors should be corrected.
Author Response
The study presented is of value regardless that the conclusion of the study sounds for quite a time in media. The scientific approach makes the results solid and may be used as trusty proof. Saying that we believe that the scientific approach was well chosen and the data presented are representative. Having said that I have the following remarks.
Patients under the study were grouped according to the Covid 19 vaccination. I am afraid it is not clearly shown what vaccines they received.
Unfortunately the breakdown of the type of vaccine used was incorrect in the original submission as it included also vaccines used after hospital admissions. We agree that a correct description of the type of vaccine used before hospital entry (the exposure of interest) is crucial. We have now corrected the calculations (lines 177-183) and added the breakdown of type of vaccine received by the participants in Table 1, including the number of the 2+ doses recipients who received heterologous vaccination. We also agree that Ad26.COV2-S vaccination is different from the others as the primary cycle consists only of one dose and therefore because of the very small number participants with history of Ad26.COV2-S (n=6) we had decided to exclude these from the study population. We have amended the Methods to clarify that we had this additional inclusion criterion in place (lines 103-105).
Nowadays it is difficult to accept the belief that one dose receivers are immunized, It would be enough to make a statement that they were at a similar risk of Covid 19 as those not vaccinated at all.
We agree with this and the information was hidden in the statistical analysis section in which we stated (lines 119-122): ‘Because the risk of the in-hospital outcomes was similar when comparing the unvaccinated with the partially vaccinated group some of the analyses were conducted using a binary exposure variable bumping together the non-vaccinated and the partially vaccinated vs. fully vaccinated’. We have now expanded the Results section to include a sentence reporting the relative hazard comparing these two groups (lines 227-230). We had already included a sentence discussing this particular finding in the original submission (lines 353-355).
However, it was not said what time after vaccination they got Covid 19. It should be discussed.
The info was missing only for the one dose group, which has been now added in Table 1.
The same is valid (elapsing time) for two doses recipients.
The median time for this group was already shown in Table 1. The figure has been slightly corrected compared to the original version which by mistake excluded the first 14 days since the date of 2nd dose.
Twenty people who received the third dose should be analyzed separately as they are at least in theory candidates for Covid19 vaccination immunity. If they failed they should be more thoroughly analyzed which should not be difficult in hospitalized patients. This analysis should include the history of illnesses especially those known to hamper immunity. In addition, the patients might be evaluated for the level of antibodies raised by vaccination.
We agree that from a clinical point of view this is an interesting group which should be looked in more details (reason for failing, etc.). However, for the purpose of this epidemiological analysis we were only interested to see whether the outcome observed in those who received at least 2 doses was driven by those who received 3 doses. Because of the small sample size of the 3 doses recipient, a formal comparison of the two groups was not possible.
Also, it is important to identify people who are primarily immunocompromised.
To describe the outcome in fragile populations was beyond the aim of our paper. Overall, the sample size of not immunocompetent participants undergoing full vaccination was small (29 patients with cancer plus 2 PLWH), too small to carry further specific analysis in this subgroup. Of course, the comparison between vaccination groups was controlled for participants’ immunocompetence via indirect adjustment for the CCI. Nevertheless, below you find the table for the primary outcome risk of death specifically adjusted for immunocompetence instead of CCI and results were similar. We have added the description of this analysis in the Methods (lines 134-136) and the table below as Supplementary material.
|
Unadjusted and adjusted relative hazards of in hospital death |
|||
|
Unadjusted HR (95% CI) |
p-value |
Adjusted* HR (95% CI) |
p-value |
Unvaccinated |
1 |
|
1 |
|
One dose |
1.25 (0.76, 2.03) |
0.377 |
0.96 (0.59, 1.58) |
0.886 |
2+ doses |
0.96 (0.65, 1.40) |
0.825 |
0.48 (0.29, 0.82) |
0.007 |
*adjusted for age, ethnicity, immunocompetence and month of enrolment |
It is important to give information about the vaccine used, some vaccines provide so-called full vaccination after one dose only.
We have now added this info (see Table 1 and Methods session, lines 101-103). Overall six participants had received Ad26.COV2-S and these were excluded.
The term fully vaccinated may be confusing I would prefer to say vaccinated with two doses and with two doses followed by a third booster shot.
We believe that the term ‘full vaccination’ has been fully explained in the Methods; further, it is in use on many other papers and the change into ‘two doses vaccination’ would burden the text.
Also, it is important to identify people who are primarily immunocompromised. The elapsing time from the second dose should be shown.
See our response to a similar comment above.
The presentation of the results seems to be quite complex which may hamper receiving the message. The following points might be considered by the Authors at revision:
- Lack of vaccination was not intentional but related to the availability of the vaccination program.
We agree with the referee that this could be one of the possible reasons for lack of vaccination and have made a little note at the beginning of the Discussion section to clarify this (lines 314-316).
- People admitted to the hospital during one year might be infected by different variants of SARS CoV2 – it should be discussed whether it was possible or not.
We had originally controlled the model for month of admission, treating it as a possible confounder for the reasons above. We have now further investigated the issue and evaluated whether calendar period (Jan-Jun vs. Jul-Dec) was an effect measure modifier for the association of interest (lines 141-143). The analysis carried no evidence for an interaction by calendar period for both the main endpoint of death (p=0.69) and mechanical ventilation or death (p=0.69). The RH from fitting a standard Cox regression model after controlling for confounding factors have been now added in Figure 4 of the revised manuscript; a sentence was also added in the discussion (lines 400-404). See also response to reviewer #3 on this point and the attacched document for Fig.4.
- Elapsing time after vaccination play a role but this is not reflected by the data presented in the table form – it should be discussed
Recent data in PLWH show that little waining in T-cell mediated response after two doses of the primary cycle of vaccine [Cicalini et al. CROI 2022, poster 291]. These new data seem to be consistent with our results shown in Table 2 of no difference in the level of protection conferred by vaccination regardless of the length of time from the second dose. We have added this speculative explanation for our findings in the revised version of the Discussion (lines 367-372).
- Risk of death and/or MV was analyzed considering confounding factors that play a significant role, it should be discussed
The threat of bias due to confounding has been carefully considered at analysis plan stage. A set of potential confounders has been identified and the assumptions leading to the identification of that set described in a direct cyclic graph (shown as Supplementary material). In addition, the concrete possibility of presence of unmeasured confounding and potential impact on the results had also been thoroughly discussed in the originally submitted version (lines 389-414).
I believe that the paper might be worthy for publication if presented in a shorter form focusing on the data contributing to the message presented in the conclusions.
Linguistical errors should be corrected.
We have revised the whole text and removed linguistical errors. We are happy to consider shortening the paper if this is strongly requested.
Reviewer 3 Report
The fast development and implementation of vaccines against current coronavirus SARS-CoV-2, which, in addition, were developed on the base of new principles, led to doubts in its effectiveness and safety. The manuscript "Real world estimate of vaccination protection in individuals hospitalized for COVID-19" is a very good example of such type of publication with detailed experimental data from large Italian hospital. But it is well-known that there were a few different coronavirus variants circulating during 2021. And therefore it was a different effectiveness of vaccines during the year. but because of the authors did not determine the variants of coronavirus, they could not separate these cases. In my opinion, they still can separate them , at least, approximately, by separating the data of the first 6 months and second 6 months of 2021 because the delta variant came to Europe in July-August. The influence of Omicron till the last two weeks of 2021 year was very small, as it could be seen from JHU morbidity/mortality data for Italy, therefore it may be not taken into account. So, in my opinion, the manuscript is of high quality and should be published but I would strongly recommend to present separately the data for 1st and 2nd halves of 2021 - at least, for mortality in connection with vaccination status. In my opinion, in this case the efficacy of vaccination could be shown substantially better in the first half of the year because vaccines protect from delta variant in much less extent than from earlier variants of coronavirus.
Author Response
The fast development and implementation of vaccines against current coronavirus SARS-CoV-2, which, in addition, were developed on the base of new principles, led to doubts in its effectiveness and safety. The manuscript "Real world estimate of vaccination protection in individuals hospitalized for COVID-19" is a very good example of such type of publication with detailed experimental data from large Italian hospital. But it is well-known that there were a few different coronavirus variants circulating during 2021. And therefore it was a different effectiveness of vaccines during the year but because of the authors did not determine the variants of coronavirus, they could not separate these cases. In my opinion, they still can separate them, at least, approximately, by separating the data of the first 6 months and second 6 months of 2021 because the delta variant came to Europe in July-August. The influence of Omicron till the last two weeks of 2021 year was very small, as it could be seen from JHU morbidity/mortality data for Italy, therefore it may be not taken into account. So, in my opinion, the manuscript is of high quality and should be published but I would strongly recommend to present separately the data for 1st and 2nd halves of 2021 - at least, for mortality in connection with vaccination status. In my opinion, in this case the efficacy of vaccination could be shown substantially better in the first half of the year because vaccines protect from delta variant in much less extent than from earlier variants of coronavirus.
We had originally controlled the model for month of admission, treating it as a possible confounder for the reasons above. We have now further investigated the issue and evaluated whether calendar period (Jan-Jun vs. Jul-Dec) was an effect measure modifier for the association of interest (lines 141-143). The analysis carried no evidence for an interaction by calendar period for both the main endpoint of death (p=0.69) and mechanical ventilation or death (p=0.69). The RH from fitting a standard Cox regression model after controlling for confounding factors have been now added in Figure 4 of the revised manuscript; a sentence was also added in the discussion (lines 400-404).
Round 2
Reviewer 2 Report
I appreciate all efforts made by the Authors to consider my remarks. Also, I understand the approach of the Authors regarding a rather conservative understanding of the term fully vaccinated. It is our duty to design vaccination programs offering a high level of immunity against emerging new variants of SARS CoV 2 Booster vaccination works.
Regarding waning of the immunity against SARS CoV 2, there is no doubt that the level of antibodies may fall beyond the protective level being much lower at least in some of the immunized individuals by 6 months after vaccination. SARS CoV 2 recognizing T cells persist well during the memory phase. However, we do not know how effective is cellular response along the elapsing time after vaccination and whether the effective protective immunity persists and if so for how long. Therefore time distance after vaccination plays a role while the risk of infection is calculated.
I appreciate considering the above remarks by the Authors.
Author Response
We thank you the reviewer for this second round of comments.
Please see our responses below.
I appreciate all efforts made by the Authors to consider my remarks. Also, I understand the approach of the Authors regarding a rather conservative understanding of the term fully vaccinated. It is our duty to design vaccination programs offering a high level of immunity against emerging new variants of SARS CoV 2 Booster vaccination works.
Data regarding the response to a 3rd and 4th dose are slowly cumulating and are at this stage uncertain. It appears that a higher proportion of the population respond to the second dose and that T-cell mediated response tend to flatten after the second dose even in fragile populations [Cicalini S et al CROI 2022]. Although partially obsolete because of the evolving situation of the pandemic (new VoC), the notion that having received the primary cycle of two mRNA vaccinations counts as ‘full vaccination’ still stands in the literature at this point in time. Because of the small proportion of participants who received the booster or indeed who were infected with Omicron our data are unable to inform future vaccination programs.
Regarding waning of the immunity against SARS CoV 2, there is no doubt that the level of antibodies may fall beyond the protective level being much lower at least in some of the immunized individuals by 6 months after vaccination. SARS CoV 2 recognizing T cells persist well during the memory phase. However, we do not know how effective is cellular response along the elapsing time after vaccination and whether the effective protective immunity persists and if so for how long. Therefore time distance after vaccination plays a role while the risk of infection is calculated.
We understand that there is concern that our current analysis reported in Table 5 (section c) comparing the effect of 2+ doses according to a fixed number of days elapsed (i.e. 180 days) from the data of the most recent dose may have been underpowered. We have now conducted an additional analysis by fitting the time since the most recent dose as a continuous variable after restricting to the 2+ doses recipients. Results are shown in the Table below. Although, in the univariable analysis there seems to be a tendency for higher risk with longer time since the most recent vaccination, this association was mostly explained by confounders (the direction of the adjusted point estimate is reversed with an HR<1). Overall, our data carried weak evidence for an association between the time since receiving the latest dose and the risk of severe clinical outcome in 2+doses recipients. We have added these new results in the revised text, added the table as Supplementary material and further modified the Discussion according to the suggestions.
|
Unadjusted and adjusted relative hazards of in hospital death |
|||
|
Unadjusted HR (95% CI) |
p-value |
Adjusted* HR (95% CI) |
p-value |
Time elapsed from second dose |
1 |
|
1 |
|
per month longer |
1.15 (0.89, 1.47) |
0.283 |
0.88 (0.56, 1.38) |
0.572 |
*adjusted for age, ethnicity, age-unadjusted CCI and month of enrolment |
